# Parents' perceptions of core outcomes in neonatal research in two Nigerian neonatal units

Sarah Kathryn Read ![ORCID],[1] Aisha Jibril,[2] Olukemi Tongo,[3] Abimbole Akindolire,[3] Isa Abdulkadir,[2] Helen Nabwera,[4] Ian Sinha,[5] Stephen Allen,[1,6] The Neonatal Nutrition Network

► Additional material is published online only. To view please visit the journal online (http://dx.doi.org/10.1136/bmjpo-2020-000669).

SKR and AJ are joint first authors.
IS and SA are joint senior authors.

For numbered affiliations see end of article.

**Correspondence to**
Sarah Kathryn Read; sarah.read@talktalk.net

## ABSTRACT

**Background** There is a scarcity of information regarding the most important outcomes for research in neonatal units in low-resource settings. Identification of important outcomes by different stakeholder groups would inform the development of a core outcome set (COS) for use in neonatal research.

**Objective** To determine the perceptions and opinions of parents of newborn babies regarding what outcomes were most important to them in order to contribute towards development of a COS for neonatal research in sub-Saharan Africa.

**Methods** Semistructured interviews were undertaken with parents, mostly mothers, of babies admitted to one neonatal unit in North central and one in Southwest Nigeria. Participants were purposively sampled to include parents of babies with common neonatal problems such as prematurity.

**Results** We conducted 31 interviews. The most frequently raised outcomes were breast feeding, good health outcomes for their baby, education, growth and financial cost. Parents placed more emphasis on quality of life and functional status than health complications.

**Conclusions** The opinions of parents need to be considered in developing a COS for neonatal research in low-resource settings. Further research should assess the opinions of families in other low-resource settings and also engage a broader range of stakeholders.

## INTRODUCTION

Although under-5 survival has improved worldwide, a child's greatest risk of dying remains in the first 28 days of life: 47% of all under-5 deaths occur in the neonatal period.[1] Effective perinatal care can have a dramatic impact on reducing the number of neonatal deaths, but little high-quality neonatal research to inform the development of interventions exists in low-resource settings.

Clinical trials are only as credible as their outcomes—the effects of an intervention.[2] Key stakeholders, including parents and clinicians, are rarely involved in outcome selection; consequently, research may not be directly relevant to them. Relevance is necessary in

### What is known about the subject?

► Variation in outcome reporting in neonatal research has limited the ability of clinical trials and meta-analyses to identify the most effective treatments.
► Little is known of the priorities of key stakeholders for neonatal research in sub-Saharan Africa.
► Parents in sub-Saharan Africa have not been involved in the selection of clinical trial outcome measures.

### What this study adds?

► Outcomes prioritised by parents relate to family outcomes and quality of life, long-term consequences of disease and short-term disease activity.
► Parents should be engaged in identifying outcomes in future research to ensure relevance to all stakeholders.

order for outcomes to influence policy and practice.[3–5] In addition, outcomes are often inconsistent, preventing the combining of results from different studies. A systematic review of studies reporting outcomes for babies in low-income and middle-income countries (LMICs) found great variation between outcomes measured and the definitions used; for example, only 60% of included studies measured survival as an outcome.[2 6]

Core outcomes sets (COSs) are groups of standardised outcomes that have been determined by key stakeholders as being the most important within a specific research field. Development and use of COS is important in developing high-quality research through standardised recording and reporting of research, reducing publication bias and facilitating systematic reviews.[2 5] COSs have already been used successfully in some areas of medical research. An example is Outcome Measures in Rheumatology which involves multiple stakeholder groups to form outcome

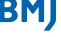

sets, leading to higher quality and more patient centred research.[7] However, few studies critically assess the selection of appropriate outcomes for research specifically for children and neonatology in particular. Variation in outcomes measured in neonatal research has limited progress; much neonatal care is inadequately evidence based as high-quality evidence is lacking and neonatal trials and meta-analyses rarely result in conclusive recommendations.[2 8]

In order to formulate COS, research and systematic reviews need to be carried out to identify important outcomes to stakeholders. Existing research on perceptions of parents and family members relating to neonatal care is focused on high-income countries. These studies reveal that important considerations for these parents include stress, neurodevelopment, survival and breast feeding.[9–11] However, research in neonatology is setting specific; research conducted in high-income countries may not be directly relevant to LMICs as the spectrum of disease differs, the same interventions are not available, cultural differences exist and adverse outcomes are much more common.[2 12] Consequently, research is needed in order to form the basis of a COS for neonatology in sub-Saharan Africa.

The Neonatal Nutrition Network (NeoNuNet; https://www.lstmed.ac.uk/nnu) was established in 2018 to contribute to improving the evidence base for neonatal care in sub-Saharan Africa where the neonatal mortality rate is 27 per 1000 compared with a global rate of 18 per 1000.[13] In Nigeria, only 39% of deliveries occur in health facilities and the neonatal mortality rate is 38 per 1000 live births.[14] Acceleration of progress in addressing neonatal mortality in Nigeria is required to meet the global sustainable development goals target of 12 per 1000 or fewer deaths by 2030 to be met.[15 16]

The aim of this study was to identify neonatal research outcomes that are most important to parents in Nigeria.

## METHODS
This study engaged parents whose newborns were admitted to one of two tertiary neonatal units in contrasting regions of Nigeria allowing comparison of two different population groups. The neonatal unit at University College Hospital, Ibadan, is located in the Southwest with a mostly urban population. The main ethnic group is the Yoruba and the dominant religion is Christianity.[17 18] Neonatal mortality is 39 per 1000 live births and 1.7% of the population live in the lowest wealth quintile for Nigeria.[19] In contrast, the neonatal unit at Ahmadu Bello University Teaching Hospita, Shika-Zaria, Kaduna, is located in North central Nigeria. The major ethnic groups are the Hausa and Fulani and the dominant religion is Islam. The population is mainly rural; 35.4% of the population live in the lowest wealth quintile and neonatal mortality rate is 44 per 1000 live births.[19]

We used semistructured interviews to achieve a balance between capturing information consistently, but also allowing families to raise issues of importance to them to avoid bias imposed by the research team. Interviews explored parents' experiences and perceptions of neonatal care[5 20] regarding the time they spent in the neonatal units[21] to formulate outcomes of importance to parents. Interviews were conducted on the neonatal units over a 4-week period in June 2019. No time limit was imposed on interviews to ensure all participants had the same opportunity to report their opinions. In line with similar previous research we aimed to conduct approximately 30 interviews across both units,[5] but acknowledged that more may be required if saturation was not reached and new themes were still emerging from the data.[22]

Participants were purposively sampled to include parents of babies with prematurity/low birth weight, birth asphyxia and congenital malformations. Parents that were regularly present on the neonatal units were selected with the assistance of clinicians, and those who agreed to share their opinions were invited to participate. Parents were excluded if their baby was deemed particularly unwell or unlikely to survive to discharge, in order to prevent unnecessary distress. Parents whose babies had been admitted for less than 3 days were also excluded as they may not have had sufficient experience of neonatal care to be able to form opinions. Parents were interviewed individually or with their partner as they preferred in a private location on the neonatal unit. Information was given to potential participants and written consent obtained. It was made clear that declining to participate in the research would not affect the clinical care being given to their baby.

An interview guide (online supplementary 1) was developed by SKR to explore the opinions of parents. SKR led the research as part of an MSc and trained non-clinical student TA and resident doctor AJ in using the interview guide. Interviews were conducted in the parents' preferred language by SKR and TA in Ibadan and AJ in Zaria. All researchers were female. Time was taken to get to know the parents prior to interview.

The interview guide was piloted among three parents. Minor revisions were made, and these pilot data were included in the final analysis. Parents were asked to reflect on what they considered important when considering whether interventions for their baby were beneficial. All interviews were recorded, transcribed and anonymised before being translated into English by Tolulope Akinrinde and AJ. Transcriptions were supplemented by notes taken during the interviews. Transcripts and notes were not returned to participants.

All parents who were invited for interview agreed to participate in the study. Thirty-three parents (30 mothers and 3 fathers) were interviewed over 31 interviews; both parents participated in two of the interviews. Sixteen interviews were conducted in Ibadan and 15 in Zaria.

### Data analysis
Analysis was iterative using an adapted framework analysis approach.[22 23] The English transcripts were read

**Table 1** Characteristics of participants

| | Location | | |
| Characteristic | Ibadan | Zaria | Total |
| --- | --- | --- | --- |
| Relationship to baby N (%) | | | |
| Mothers | 15 (45) | 15 (45) | **30 (91)** |
| Fathers | 3 (9) | 0 (0) | **3 (9)** |
| **Total** | **18 (55)** | **15 (45)** | **33 (100)** |
| Sex of baby N (%) | | | |
| Male | 6 (19) | 11 (34) | **17 (53)** |
| Female | 11 (34) | 4 (13) | **15 (47)** |
| **Total** | **17 (53)** | **15 (47)** | **32 (100)*** |
| Maternal education status N (%) | | | |
| University graduate or equivalent | 7 (23) | 6 (19) | **13 (42)** |
| Senior secondary school certificate holders who have teaching or other professional training | 3 (10) | 1 (3) | **4 (13)** |
| Senior secondary school certificate holders or grade II teachers certificate holders equivalent | 4 (13) | 2 (6) | **6 (19)** |
| JSS3† or primary six certificate | 2 (6) | 5 (16) | **7 (23)** |
| Those who can either just read or write or are illiterate | 0 (0) | 1 (3) | **1 (3)** |
| **Total** | **16 (52)** | **15 (48)** | **31 (100)** |
| Maternal occupation N (%) | | | |
| Senior public servants, professionals, managers, large scale traders, businessmen and contractors | 1 (3) | 0 (0) | **1 (3)** |
| Intermediate grade public servants and senior school teachers | 2 (6) | 1 (3) | **3 (10)** |
| Junior school teachers, drivers and artisans | 6 (19) | 2 (6) | **8 (26)** |
| Petty traders, labourers, messengers and similar grades | 4 (13) | 0 (0) | **4 (13)** |
| Unemployed, students, full-time housewives and subsistence farmers | 3 (10) | 12 (39) | **15 (48)** |
| **Total** | **16 (52)** | **15 (48)** | **31 (100)** |

*One mother had twins on the neonatal unit.
†Junior secondary school examination.

## RESULTS

Participant characteristics are shown in table 1. Mean maternal age was 29 years (range 18–44 years). Across both neonatal units, 13 out of 31 mothers had been to university; 7 in Ibadan and 6 in Zaria. However, 15 were either unemployed or full-time housewives but there was significant variation between the two units—3 in Ibadan but 12 in Zaria. Most participants in Ibadan were Christian and all participants in Zaria were Muslim. Two parents had previously had babies who were admitted to the same neonatal unit.

Median gestation at delivery was 35 weeks (range 26–40 weeks) and median birth weight was 1.75 kg (range 0.75–3.90 kg). The mean duration of admission prior to interview was 6 days (range 3–17 days). These factors were similar in both neonatal units (data not shown). The main reason for admission of the babies are shown in table 2. Prematurity/low birth weight was the primary indication for admission in a greater proportion of babies in Ibadan than Zaria.

Interviews lasted between 10 and 30 min. Parents identified 22 broad outcomes. Three were only mentioned in one interview and so were excluded (passing of stool, appropriate sleeping, normal temperature). The most commonly identified outcomes are presented with illustrative quotes in table 3: breast feeding, good

multiple times by SKR and recurring themes noted. These themes were then coded before related codes were grouped together to form broad themes. As each new transcript was read, it was compared with those coded previously to ensure consistency. Rather than using a coding framework, broad themes were rephrased as outcomes and discussed at a NeoNuNet meeting comprising of neonatal clinical leads and the research team. To prioritise outcomes, those mentioned only once were discarded and those remaining were listed in order of how commonly they were discussed by the participants.

The Consolidated criteria for Reporting Qualitative research reporting guidelines were used.[24]

**Table 2** Main clinical indication for admission of the babies

| Location | Very premature (gestation <32 weeks) or very low birth weight (<1500 g) N (%) | Asphyxia N (%) | Congenital malformation* N (%) | Other† N (%) | Total N (%) |
|---|---|---|---|---|---|
| Ibadan | 10 (31) | 1 (3) | 1 (3) | 5 (16) | **17 (53)** |
| Zaria | 2 (6) | 4 (13) | 2 (6) | 7 (22) | **15 (47)** |
| Total | **12 (38)** | **5 (16)** | **3 (9)** | **12 (38)** | **32 (100)** |

*Posterior urethral valves, cystic hygroma, spina bifida.
†Gestational age 32–36 weeks, low birth weight <2500 g, jaundice, sepsis.

health outcomes for baby, education, growth, religious factors and financial cost. Table 4 shows the full list of outcomes identified by parents, ranked by how often they were mentioned. When grouped into broad outcome domains, it was clear parents were concerned with family outcomes and quality of life, long-term consequences of disease and short-term illnesses.

Across both units, the main concerns of parents related to breast feeding and good health outcomes for the baby. More parents in Ibadan mentioned concerns regarding educational ability, growth and length of hospital stay

than parents in Zaria. Both groups of parents frequently mentioned the importance of religion with regard to outcomes for their baby and their faith they have in God.

No new themes were identified by parents in the final six interviews; therefore, we considered that saturation had been achieved with the first 25 interviews.

## DISCUSSION

This study provides insights into outcomes of neonatal care that are important to parents in two tertiary neonatal

**Table 3** Illustrative quotes for the top six prioritised outcomes

| Outcome | Illustrative quotes |
|---|---|
| Breast feeding | 'I feel that because they're [healthcare workers] not allowing me to breastfeed I feel that baby is not well yet' (mother, Ibadan).<br>'If the baby can breastfeed directly by herself then it shows that she has improved more than she was' (mother, Ibadan).<br>'What will fill my joy is(…)breastfeeding the baby, instead of looking at him from afar' (mother, Ibadan)<br>'I want the baby to take more of the breast milk(…)I want her to be able to feed' (mother, Zaria). |
| Good health outcomes for baby | 'My major concern is for my baby to be fine' (mother, Ibadan).<br>'I want to see them [babies] perfectly fine. Stronger, healthy' (mother, Ibadan).<br>'The baby should be well and healthy(…)he should be healthy in that he should come back to normalcy' (mother, Zaria).<br>'What bothers me most is my baby to be healthy' (mother, Zaria). |
| Education | 'I want him [baby] to be very intelligent. I'm really concerned about education' (mother, Ibadan).<br>'Most important(for baby's future)? Education(…)he must do well at school' (mother, Ibadan)<br>'I want my baby to have good education(…)that is what I'm most concerned' (mother, Ibadan)<br>'I pray he [baby] will be able to read well' (mother, Zaria). |
| Growth | 'We want the baby to remain getting big. Remain getting big, not be getting small' (father, Ibadan).<br>'I want her [baby] to grow very big, carriable, workable for me to be happy' (mother, Ibadan).<br>'Her(baby's)weight is important to measure(…)she was born small and has put on weight and it shows she is getting better' (mother, Ibadan).<br>'I am concerned that the baby should grow well' (mother, Zaria). |
| Religious factors | 'The future is in God's hands, there's nothing to worry about' (mother, Ibadan).<br>'I pray to God, may God take care of my baby' (father, Ibadan).<br>'God will provide. Because he gave them [babies] to me and so I know he will provide. I know God will look after them' (mother, Ibadan).<br>'I don't have any worry because it is God that gives health' (mother, Zaria) |
| Financial cost | 'The difficult things have been basically financial' (mother, Ibadan).<br>'I have worries for money now. Because I don't know how much it's going to cost me' (mother, Ibadan).<br>'Financial cost really is an issue because I had caesarean section, so we spent a lot of money so truly we have financial issues' (mother, Zaria).<br>'(I)am worried about getting money to be able to make ends meet' (mother, Zaria). |

**Table 4** Outcomes identified during interviews with parents according to site and ranked by overall frequency

| Rank | Outcome domain | Total no of times identified N (%) | Times identified in Ibadan N (%) | Times identified in Zaria N (%) |
|---|---|---|---|---|
| 1 | Breast feeding | 22/31 (71) | 11/16 (69) | 11/15 (73) |
| 2 | Good health outcomes for baby | 20/31 (65) | 8/16 (50) | 12/15 (80) |
| 3 | Education | 16/31 (52) | 10/16 (63) | 6/15 (40) |
| 4 | Growth | 16/31 (52) | 11/16 (69) | 5/15 (33) |
| 5 | Religious factors* | 15/31 (48) | 7/16 (44) | 8/15 (53) |
| 6 | Financial cost | 11/31 (35) | 7/16 (44) | 4/15 (27) |
| 7 | Length of hospital stay | 9/31 (29) | 8/16 (50) | 1/15 (7) |
| 8 | Survival to discharge | 8/31 (26) | 5/16 (31) | 3/15 (20) |
| 9 | Jobs or future achievements | 7/31 (23) | 6/16 (38) | 1/15 (7) |
| 10 | Coming off of oxygen | 7/31 (23) | 5/16 (31) | 2/15 (13) |
| 11 | Alertness of baby | 6/31 (19) | 2/16 (13) | 4/15 (27) |
| 12 | Breathing | 5/31 (16) | 2/16 (13) | 3/15 (20) |
| 13 | Baby able to cry | 5/31 (16) | 0 (0) | 5/15 (33) |
| 14 | Jaundice | 5/31 (16) | 1/16 (6) | 4/15 (27) |
| 15 | Phototherapy | 4/31 (13) | 4/16 (25) | 0 (0) |
| 16 | Able to hold baby | 3/31 (10) | 3/16 (19) | 0 (0) |
| 17 | Medications | 3/31 (10) | 2/16 (13) | 1/15 (7) |
| 18 | Development | 2/31 (6) | 1/16 (6) | 1/15 (7) |
| 19 | Survival to adulthood | 2/31 (6) | 2/16 (13) | 0 (0) |

*Refers to the influence of religion on outcomes for babies, beliefs of the parents that outcomes are in the hands of God.

units in Nigeria. These findings can be used to inform the development of COS for neonatal research in sub-Saharan Africa.

It is clear that parents were concerned with issues they were facing as part of their lived experience of having a baby on the neonatal unit. The most frequent outcome identified by parents was breast feeding; the ability to breastfeed was felt by most to be a sign of improvement in their baby. Growth was also raised as a key concern. The link between good feeding and growth was raised by parents and is well known.[25] Parents also identified long-term outcomes as important; they specified that education of their child in particular was important to them, as were future jobs and achievements. However, this is rarely identified as a concern in existing research.[5] Although challenging, greater emphasis on long-term follow-up would enhance the relevance of future research to parents. The financial cost to the family of the current admission was also frequently mentioned by parents but again, do not appear in research from high-income countries. This is not surprising as the relative expenses associated with healthcare are much greater in Nigeria than they are in high-income settings, highlighting the importance of context-specific outcome sets.[26 27]

Mortality is a key measure of neonatal outcome[6] and so it was surprising that mortality was rarely mentioned in interviews. This may suggest a fatalistic attitude towards the death of a newborn due to the high rates of neonatal mortality, as most parents interviewed had low birth weight infants whose risk of mortality would be high. However, it is important to put this in the perspective that babies who were severely unwell or not likely to survive were excluded from this study. Religious beliefs were mentioned in about half of interviews, suggesting that they may be valuable coping strategies in an environment where adverse outcomes for newborns are common. There is a lack of research on the impact of religious beliefs on neonatal care although their impact on uptake of maternal health services has been studied.[28] Religious beliefs were mentioned frequently across both locations in our study, highlighting that religious beliefs may affect attitudes towards neonatal care in both Christian and Muslim populations of Nigeria. Further research would help delineate how these values shape attitudes and priorities of parents, and also healthcare workers, in relation to newborn care.

### Strengths and limitations

Despite the overall similarity of results from each unit, researchers in other low-resource settings should interpret the transferability of our findings to their own contexts. We included two tertiary units located in major cities in two distinct regions of Nigeria; findings may not

be transferable to other tertiary units or primary and secondary centres that care for neonates in these settings.

We excluded particularly sick babies and those thought unlikely to survive. Therefore, our findings are not directly relevant to this group. Most babies had been admitted for less than a week when their parents were interviewed. As such, parents may still be coming to terms with preterm delivery and/or having a sick baby and may not have had enough time to consider and formulate their longer-term concerns. Parents' opinions may change as hospital admission progresses. This should be considered in identifying important outcomes for interventions occurring towards the end of admission. Another limitation of our study was that few fathers were interviewed, limiting the results from these important stakeholders. Although we included parents of babies with common neonatal problems, their views may not be representative of parents whose babies have other conditions.

Parents may not have felt in a position to truthfully disclose concerns due to their vulnerable position as parents of sick babies receiving care on a neonatal unit, or their opinions may have been influenced by local health workers. Finally, all babies in this study required hospitalisation and so the results may not be generalisable to all newborn babies, even in settings with high neonatal mortality.

Lack of parental involvement in this study design is a major limitation to this study. However, our findings from this study can help guide patient and public involvement for further research in the development of COS.

In order to ensure that COS in neonatal research would be adopted by researchers, further research should engage a wider group of stakeholders including regulators, clinical trial authorities and policy-makers. We have identified outcomes of importance to parents; further research is required to standardise and validate the tools and measures used to evaluate these outcomes and the time points at which they are measured. This study highlights the need for development of COS in neonatology in order to improve outcomes for preterm infants and that future research in this field should consider the needs of families when prioritising research outcomes.

## CONCLUSIONS

This study has identified outcomes of particular importance to parents of babies on tertiary neonatal units in two regions of Nigeria; breastfeeding, good health outcomes for baby, education, growth and financial cost. Our findings can inform further research in COS for neonatal trials in sub-Saharan Africa.

**Author affiliations**
[1]Department of Clinical Sciences, Liverpool School of Tropical Medicine, Liverpool, UK
[2]Department of Paediatrics, Faculty of Clinical Sciences, College of Medical Sciences, Ahmadu Bello University, Zaria, Kaduna, Nigeria
[3]Department of Paediatrics, College of Medicine, University of Ibadan, Ibadan, Oyo, Nigeria
[4]International Public Health, Liverpool School of Tropical Medicine, Liverpool, UK
[5]Respiratory Medicine, Alder Hey Children's NHS Foundation Trust, Liverpool, UK
[6]Department of Gastroenterology, Alder Hey Children's NHS Foundation Trust, Liverpool, UK

**Acknowledgements** We are grateful to the parents of babies admitted to the neonatal units in Ibadan and Zaria for contributing to our research. Thank you to Tolulope Akinrinde for her work as translator in Ibadan.

**Collaborators** Neonatal Nutrition Network members: Olusegun Akinyinka (College of Medicine, University of Ibadan, Nigeria); Dominic Umoru (Maitama District Hospital, Abuja, Nigeria); Chinyere Ezeaka (Lagos University Teaching Hospital, Nigeria); Ireti Fajolu (Lagos University Teaching Hospital, Nigeria); Beatrice Ezenwa (Lagos University Teaching Hospital, Nigeria); Zainab Imam (Massey St. Children's Hospital, Lagos, Nigeria); Martha Mwangome (KEMRI Wellcome Trust Research Programme, Kilifi, Kenya); Alison Talbert (KEMRI Wellcome Trust Research Programme, Kilifi, Kenya); Pauline Andang'o (Jaramogi Oginga Odinga Teaching and Referral Hospital, Kisumu, Kenya); Walter Otieno (Jaramogi Oginga Odinga Teaching and Referral Hospital, Kisumu, Kenya; Maseno University, Kisumu, Kenya); Grace Nalwa (Jaramogi Oginga Odinga Teaching and Referral Hospital, Kisumu, Kenya); Janneke van de Wijgert (University of Liverpool, Liverpool, UK); Melissa Gladstone (University of Liverpool, Liverpool, UK); Kevin Mortimer (Liverpool School of Tropical Medicine, Liverpool, UK); Graham Devereux (Liverpool School of Tropical Medicine, Liverpool, UK); Ismaela Abubakar (Liverpool School of Tropical Medicine, Liverpool, UK); Nicholas Embleton (Newcastle University, Newcastle, UK)

**Contributors** IS and SA conceived the project. SKR and AJ undertook the data collection in Nigeria. SKR performed the data analysis and wrote the first draft of the manuscript. All authors contributed to project design and data interpretation and approved the final manuscript.

**Funding** This project was completed as part of the Neonatal Nutrition Network, funded by a grant from the MRC Confidence in Global Nutrition and Health Research scheme (grant reference MC_PC_MR/R019789/1).

**Competing interests** None declared.

**Patient and public involvement** Patients and/or the public were not involved in the design, or conduct, or reporting, or dissemination plans of this research.

**Patient consent for publication** Not required.

**Ethics approval** Ethical approval was obtained from the Liverpool School of Tropical Medicine Research and Ethics Committee and the ethics committees at University College Hospital, Ibadan, and Ahmadu Bello University Teaching Hospital, Zaria.

**Provenance and peer review** Not commissioned; externally peer reviewed.

**Data availability statement** Data are available on reasonable request. Anonymised participant data along with original transcripts and coding is available from the research team on reasonable request.

**ORCID iD**
Sarah Kathryn Read http://orcid.org/0000-0003-2328-0992

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
