## [Reviewer comments · BMJ Paediatrics Open]

ARTICLE DETAILS

TITLE (PROVISIONAL)	Parents' perceptions of core outcomes in neonatal research in two Nigerian neonatal units
AUTHORS	Read, Sarah; Jibril, Aisha; Tongo, Olukemi; Akindolire, Abimbole; Abdulkadir, Isa; Nabwera, Helen; Sinha, Ian; Allen, Stephen

VERSION 1 – REVIEW

REVIEWER	Reviewer name: Mandy Daly Institution and Country: Irish Neonatal Health Alliance, Ireland Competing interests: Non
REVIEW RETURNED	03-Mar-2020

GENERAL COMMENTS	It struck me as ironic that a study which reaches the conclusion that families should be included in the development of a future COS did not include a PPI in the study design and throughout the process. PPI input may well have questioned the purpose of this study as its findings arise from "semi-structured" interviews (which could lead to bias due to the lack of a definitive structure) with a very small number of participants from 2 city based tertiary units (limiting the transferability of the findings). As the interviews ranged in time frame from 10 mins - 30 mins it is difficult to understand if all participants had "the same" conversation with their respective interviewer. A PPI might have pointed out the fact that interviewing families who's infant is only in the Neonatal Unit a short time is not likely to elicit opinions similar to families who's infants have been in the unit for a considerable length of time (these families are better placed to comment on long term concerns). The study findings (relatively broad in nature) reflect what many COS developers would include anyway when developing a COS (with PPI input) regarding the most important outcomes for families. In the current COS development climate the inclusion of parents/families is considered obligatory and necessary hence I find this study's conclusion that the opinions of parents need to be considered in developing a COS for neonatal research in low-resource settings simply a reiteration of a criteria that is already available in the COS development domain.
--

REVIEWER	Reviewer name: Julia Petty Institution and Country: University of Hertfordshire, Hatfield, UK Competing interests: Nil
REVIEW RETURNED	03-Mar-2020

GENERAL COMMENTS	Thank you - a well-written and comprehensive paper. The inclusion of parents involvement is important and a welcome feature that is so often omitted. I have just made 4 suggestions: 1- Page 10, 1st line under Data Analysis sub-title- re; your mode of analysis, more information (2-3 sentences) is required relating to this - what is this and how was it 'adapted'. How did the themes 'emerge'? 2- Page 10- sub-title Patient and Public Involvement - Is this section
--

	really required? it does not fit with the flow of the text and sounds very negative. You are involving and interviewing parents so surely this counts as 'patient involvement'beit indirectly. 3- Results- 1st section - This 1st paragraph / section is not results. This part should be moved to be in the discussion of the methods and how the research was undertaken. 4- Discussion - More detail (just brief) could be included on the different religions for the two hospitals and how this may have influenced the findings / themes. Otherwise, a worthwhile topic and paper.
--	--

REVIEWER	Reviewer name: Wally Carlo Institution and Country: University of Alabama at Birmingham United States of America Competing interests: None
REVIEW RETURNED	24-Mar-2020

GENERAL COMMENTS	31 semi-structures interviews with parents of babies from two neonatal units in Nigeria were conducted in a cross-sectional study to determine the perceptions and opinions of outcomes most important to them in order to contribute towards development of a set of core outcomes for neonatal research in sub-Saharan Africa. Importantly, this is a small study conducted in only two neonatal units in Nigeria. Other stakeholders were not included. Some of the parents' perceptions and opinions may be affected by local health workers so the use of only two neonatal units limits generalizability. Abstract The Conclusions are too strong as they make recommendation of study design but these recommendations are based on few parents in just two neonatal units in Nigeria. The authors do not represent a national or international organization to justify sweeping recommendations. Furthermore, stakeholders other than parents are not included. There are concerns about generalizability. The Conclusions should be limited to interpretation and summary of the results of the study rather than make blanket recommendations. Methods It is important to note that parents of unwell babies as well as parents of babies not likely to survive were excluded. This exclusion may be better from the local implementation of research but in country with such high risk for neonatal mortality, this is an important limitation as parents of the babies most likely to die were systematically excluded. Results The results differed substantially between the two units. This raises concerns about generalizability to other neonatal units and in particular, neonatal units in other countries and cultures. Discussion It is stated that it was surprising that mortality was not a main concern but it is important to put it in the perspective that infants at the highest risk of death were systematically excluded.
--

VERSION 1 – AUTHOR RESPONSE

Reviewer 1

- It struck me as ironic that a study which reaches the conclusion that families should be included in the development of a future COS did not include a PPI in the study design and throughout the process.

Response: The reviewer is correct to note that parents were not involved in the design of this study and we have highlighted this as a major limitation (page 13, lines 39-44) However, consistent with reviewer 2's comments regarding 'patient involvement', we consider that our findings will help guide the engagement of PPI in further research for developing COS.

"Lack of parental involvement in this study design is a major limitation to this study. However, our findings from this study can help guide patient and public involvement (PPI) for further research in the development of core outcome sets."

- PPI input may well have questioned the purpose of this study as its findings arise from "semi-structured" interviews (which could lead to bias due to the lack of a definitive structure) with a very small number of participants from 2 city based tertiary units (limiting the transferability of the findings).

Response: We chose semi-structured interviews to achieve a balance between capturing uniform information across the different participants but also to allow families to raise issues of importance to them to avoid bias imposed by the research team (clarified on page 8, lines 18-22). We have emphasised in the section on strengths and limitations (page 12, lines 50-52) that the transferability of our findings to other settings, even with LMICs, may be limited.

"We used semi-structured interviews to achieve a balance between capturing information consistently, but also allowing families to raise issues of importance to them to avoid bias imposed by the research team."

"Despite the overall similarity of results from each unit, researchers in other low-resource settings should interpret the transferability of our findings to their own contexts."

- As the interviews ranged in time frame from 10 mins - 30 mins it is difficult to understand if all participants had "the same" conversation with their respective interviewer.

Response: Although the duration of interviews varied, there was no time limit imposed. As a result, we consider that all participants had the same opportunity to report their opinions and, therefore, our methodology was consistent across interviews (clarified on page 8, lines 29-31).

"No time limit was imposed on interviews to ensure all participants had the same opportunity to report their opinions."

- A PPI might have pointed out the fact that interviewing families who's infant is only in the Neonatal Unit a short time is not likely to elicit opinions similar to families who's infants have been in the unit for a considerable length of time (these families are better placed to comment on long term concerns).

Response: Thank you for raising this important point. Although we excluded families whose infant had been admitted for <3 days, we were keen to include families whose infant had spent different periods of time on the unit as COS need to be relevant to interventions in the first few months of admission (e.g. starting enteral feeds) as well as longer-term (e.g. KMC). We agree that families' opinions may differ according to duration of admission and we have noted this (page 13, lines 12-16).

"Parents' opinions may change as hospital admission progresses. This should be considered in identifying important outcomes for interventions occurring towards the end of admission."

- The study findings (relatively broad in nature) reflect what many COS developers would include anyway when developing a COS (with PPI input) regarding the most important outcomes for families. In the current COS development climate the inclusion of parents/families is considered obligatory and necessary hence I find this study's conclusion that the opinions of parents need to be considered in developing a COS for neonatal research in low-resource settings simply a reiteration of a criteria that is already available in the COS development domain.

Response: We agree with the reviewer but feel that our study highlights the importance of including parents and family members in developing COS especially given the current lack of information on opinions of parents and family members from low-income countries. We hope that this study can be used to aid further research into COS in these settings and highlights some of the reasons why parental involvement is key.

Reviewer 2

- Page 10, 1st line under Data Analysis sub-title- re; your mode of analysis, more information (2-3 sentences) is required relating to this - what is this and how was it 'adapted'. How did the themes 'emerge'?

Response: We thank the reviewer for the comments to expand on our mode of analysis. More information has been provided (page 9, lines 50-60) which we believe clarifies what the analysis process involved. A framework analysis approach was used but instead of putting identified themes and codes into a framework, these were rephrased as outcomes for classification.

“The English transcripts were read multiple times by SR and recurring themes noted. These themes were then coded before related codes were grouped together to form broad themes. As each new transcript was read, it was compared to those coded previously to ensure consistency. Rather than using a coding framework, broad themes were rephrased as outcomes and discussed at a NeoNuNet meeting comprising of neonatal clinical leads and the research team.”

- Page 10- sub-title Patient and Public Involvement - Is this section really required? it does not fit with the flow of the text and sounds very negative. You are involving and interviewing parents so surely this counts as 'patient involvement' be it indirectly.

Response: We have removed the specific section on PPI and, as stated above, have clarified parental involvement (page 13, lines 39-44).

- Results- 1st section - This 1st paragraph / section is not results. This part should be moved to be in the discussion of the methods and how the research was undertaken.

Response: We thank the reviewer for highlighting this to us. We have now moved this paragraph into the methods section (page 9, lines 37-43).

- Discussion - More detail (just brief) could be included on the different religions for the two hospitals and how this may have influenced the findings / themes.

Response: We have emphasised the potential importance of religious beliefs in the discussion section (page 12, lines 35-37) and have already highlighted the need for further research.

“Religious beliefs were mentioned frequently across both locations in our study, highlighting that religious beliefs may affect attitudes towards neonatal care in both Christian and Muslim populations of Nigeria.”

Reviewer 3

- 31 semi-structured interviews with parents of babies from two neonatal units in Nigeria were conducted in a cross-sectional study to determine the perceptions and opinions of outcomes most important to them in order to contribute towards development of a set of core outcomes for neonatal research in sub-Saharan Africa. Importantly, this is a small study conducted in only two neonatal units in Nigeria. Other stakeholders were not included. Some of the parents' perceptions and opinions may be affected by local health workers so the use of only two neonatal units limits generalizability.

Response: We thank the reviewer for raising further limitations to our study and we have added this information to the strengths and limitations section (page 12, lines 52-56 and page 13, lines 26-31). Although there are significant limitations to the transferability of our findings, we still feel that it raises important points that should be considered in future COS research.

“We included two tertiary units located in major cities in two distinct regions of Nigeria; findings may not be transferable to other tertiary units or primary and secondary centres that care for neonates in these settings.”

“Parents may not have felt in a position to truthfully disclose concerns due to their vulnerable position as parents of sick babies receiving care on a neonatal unit, or their opinions may have been influenced by local health workers.”

- Abstract: The Conclusions are too strong as they make recommendation of study design, but these recommendations are based on few parents in just two neonatal units in Nigeria. The authors do not represent a national or international organization to justify sweeping recommendations. Furthermore, stakeholders other than parents are not included. There are concerns about generalizability. The Conclusions should be limited to interpretation and summary of the results of the study rather than make blanket recommendations.

Response: We have modified our conclusions and incorporated these comments (Abstract – page 4, lines 54-58; Conclusion – page 14, lines 14-20).

“Further research should assess the opinions of families in other low-resource settings and also engage a broader range of stakeholders.”

- Methods: It is important to note that parents of unwell babies as well as parents of babies not likely to survive were excluded. This exclusion may be better from the local implementation of research but in country with such high risk for neonatal mortality, this is an important limitation as parents of the babies most likely to die were systematically excluded.

Response: Once again, we thank the reviewer for raising another limitation of this study. Although all of the infants in the study were sufficiently unwell to require hospital admission and mortality in this population is very high, we have emphasised that our findings are not directly transferable to severely unwell infants (page 13, lines 3-8 and lines 31-35).

“We excluded particularly sick babies and those thought unlikely to survive. Therefore, our findings are not directly relevant to this group.”

“All babies in this study required hospitalisation and so the results may not be generalisable to all newborn babies, even in settings with high neonatal mortality.”

- Results: The results differed substantially between the two units. This raises concerns about generalizability to other neonatal units and in particular, neonatal units in other countries and cultures.

Response: We have also highlighted this in our strengths and limitations section (page 12, lines 50-52) and feel this is clear in our results tables for people to interpret. Concerns about transferability have been emphasised as we feel this has been raised as a key limitation to this study.

- Discussion: It is stated that it was surprising that mortality was not a main concern, but it is important to put it in the perspective that infants at the highest risk of death were systematically excluded.

Response: As stated above, we have clarified the exclusion of sick infants as a limitation and we have emphasised this in the discussion section (page 12, lines 24-29) and strengths and limitations (page 13, lines 3-8) for further clarity and to aid interpretation of our findings.

“However, it is important to put this in the perspective that babies who were severely unwell or not likely to survive were excluded from this study.”

VERSION 2 – REVIEW

REVIEWER	Reviewer name: Mandy Daly Institution and Country: Irish Neonatal Health Alliance Competing interests: NIL
REVIEW RETURNED	23-Apr-2020
GENERAL COMMENTS	Thank you for taking on board the suggestions from previous reviews.
REVIEWER	Reviewer name: Julia Petty Institution and Country: Univ Hertfordshire Competing interests: None
REVIEW RETURNED	03-May-2020
GENERAL COMMENTS	Thank you. Your responses to the reviewers' suggestions and recommendations have been addressed adequately in my view and the amended manuscript reflects the necessary changes
REVIEWER	Reviewer name: Waldemar A Carlo Institution and Country: University of Alabama at Birmingham, USA Competing interests: None
REVIEW RETURNED	05-May-2020
GENERAL COMMENTS	None. Authors addressed all concerns.